# Improving Continual Learning by Accurate Gradient Reconstructions of the Past

**Erik Daxberger**                                                      *ead54@cam.ac.uk*
*University of Cambridge & MPI for Intelligent Systems, Tübingen*

**Siddharth Swaroop**                                              *siddharth@seas.harvard.edu*
*Harvard University*

**Kazuki Osawa**                                                  *kazukiosawa@google.com*
*Google DeepMind*

**Rio Yokota**                                                 *rioyokota@gsic.titech.ac.jp*
*Tokyo Institute of Technology*

**Richard E. Turner**                                                    *ret26@cam.ac.uk*
*University of Cambridge*

**José Miguel Hernández-Lobato**                                      *jmh233@cam.ac.uk*
*University of Cambridge*

**Mohammad Emtiyaz Khan**                                          *emtiyaz.khan@riken.jp*
*RIKEN Center for AI Project*

**Reviewed on OpenReview:** *https://openreview.net/forum?id=b1fpfCjja1*

## Abstract

Weight-regularization and experience replay are two popular continual-learning strategies with complementary strengths: while weight-regularization requires less memory, replay can more accurately mimic batch training. How can we combine them to get better methods? Despite the simplicity of the question, little is known or done to optimally combine these approaches. In this paper, we present such a method by using a recently proposed principle of adaptation that relies on a faithful reconstruction of the gradients of the past data. Using this principle, we design a prior which combines two types of replay methods with a quadratic weight-regularizer and achieves better gradient reconstructions. The combination improves performance on standard task-incremental continual learning benchmarks such as Split-CIFAR, SplitTinyImageNet, and ImageNet-1000, achieving $> 80\%$ of the batch performance by simply utilizing a memory of $< 10\%$ of the past data. Our work shows that a good combination of the two strategies can be very effective in reducing forgetting.

## 1 Introduction

Continual learning (Parisi et al., 2019) aims for accurate incremental training over a large number of individual tasks/examples. This can potentially reduce the frequency of retraining in deep learning, making algorithms easier to use and deploy, while also reducing their environmental impact (Diethe et al., 2019; Paleyes et al., 2020). The main challenge in continual learning is to remember past knowledge and reuse it to continue to adapt to new data. This can be difficult because the future is unknown and can interfere with past knowledge (Sutton, 1986; Mermillod et al., 2013; Kirkpatrick et al., 2017). Performance, therefore, heavily depends on the strategies used to represent and reuse past knowledge.

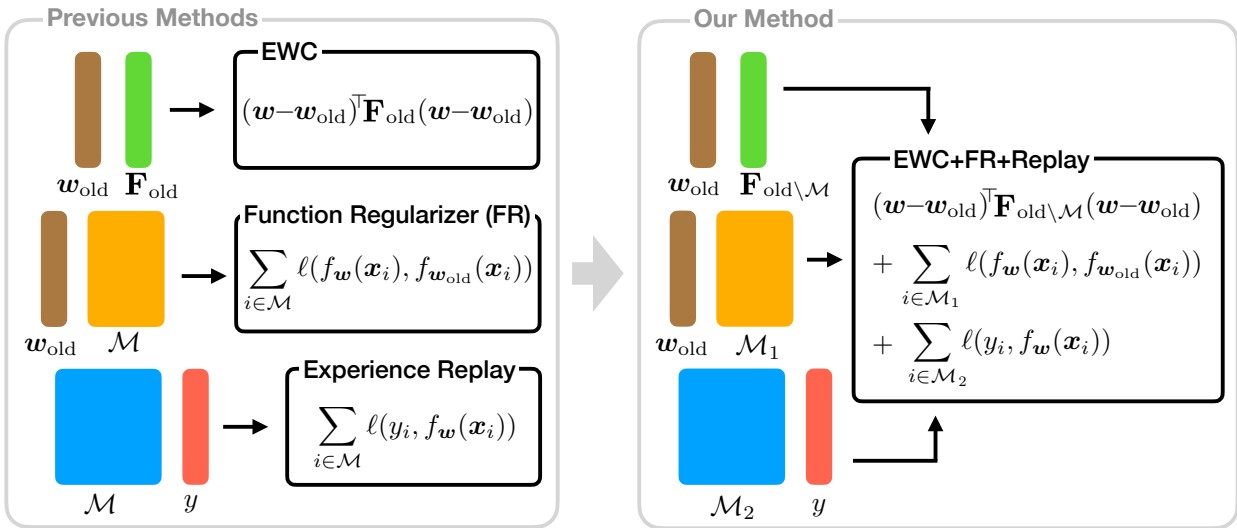

Figure 1: Using the principle of gradient reconstructions, we design a new prior (right) to combine different types of regularization and memory-based methods (left). EWC uses a quadratic regularizer based on the old weight-vector $\boldsymbol{w}_{\text{old}}$ and its importance $\mathbf{F}_{\text{old}}$, while experience replay uses a memory $\mathcal{M}$ of past examples along with their labels. Function regularization (FR) is similar but often does not use labels $y$. Our method combines EWC, FR, and replay by using two memory sets, $\mathcal{M}_1$ and $\mathcal{M}_2$. The notable differences are that our method's importance matrix excludes the examples in $\mathcal{M} = \mathcal{M}_1 \cup \mathcal{M}_2$, and that experience replay is applied *only* to the second memory set $\mathcal{M}_2$. The examples in $\mathcal{M}_1$ do not require labels, and can be compressed to only keep a small set of representative inputs that may not be part of the old data; see Section 5 for details.

Two popular strategies of knowledge reuse are based on weight-regularization and experience replay. The two strategies have complementary strengths. For example, the well-known Elastic-Weight Consolidation (EWC) (Kirkpatrick et al., 2017), which regularizes the new weight-vector to keep it close to the old one, requires a fixed memory size to store the weight vector and its importance. A variety of other such regularizers have been proposed (Schwarz et al., 2018; Zenke et al., 2017; Li & Hoiem, 2017; Nguyen et al., 2018). Experience replay (Robins, 1995; Shin et al., 2017), on the other hand, often increases memory as learning progresses. Memory cost here can be substantial, but it can boost accuracy if the memory represents the past well. Clearly, combining the two approaches can strike a good balance between performance and memory size. At present, little has been done to find principled ways to combine them. Some works have used knowledge distillation (Rebuffi et al., 2017; Buzzega et al., 2020) or functional regularization (Titsias et al., 2020; Pan et al., 2020), where predictions evaluated at the examples in memory are regularized. Such attempts are promising, but it is not clear why the specific choices of regularizers and memory work well, and whether there are better choices that lead to further improvements.

In this paper, we provide a new approach to combine and improve weight-regularization and experience replay. The approach is based on a recently proposed principle of adaptation, where a prior called the Knowledge-adaptation prior (K-prior) is used to reconstruct the gradients of the past training objective (Khan & Swaroop, 2021). We use this as a guideline to design better priors for continual learning. Khan & Swaroop (2021) consider only one-task adaptation, and relied on a simple quadratic regularizer without any experience replay. We extend their method to multiple tasks and use it to combine a weight-regularizer, a functional regularizer, and experience replay (Fig. 1). Each piece contributes in a complementary way to the reduction of a different type of error, and the combination overall gives lower gradient-reconstruction error than each individual component. This leads to consistent improvements on standard benchmarks for multi-task image classification in task-incremental continual learning, such as Split-CIFAR, Split-TinyImageNet, and ImageNet-1000, across various memory budgets from small to large sizes. On these benchmarks, our method can achieve $> 80\%$ of the batch performance by simply utilizing a memory of $< 10\%$ of the past data. Our approach yields better strategies than the current heuristics used in the literature.

## 2 Related work

**Continual learning paradigms.** Existing CL methods can be classified into three complementary paradigms or combinations thereof: 1) *weight-regularization methods* regularize parameter updates by penalizing changes in 'important' parameters for previous tasks, 2) *memory-replay methods* rehearse some past data or pseudo-data, and 3) *architecture-based methods* change the model architecture or mask model weights (Parisi et al., 2019; Swaroop, 2022). Here, we focus on the first two CL paradigms. Weight-regularization approaches include Elastic Weight Consolidation (EWC) (Kirkpatrick et al., 2017), Online EWC (Schwarz et al., 2018), Synaptic Intelligence (SI) (Zenke et al., 2017), Learning without Forgetting (LwF) (Li & Hoiem, 2017), and Variational Continual Learning (VCL) (Nguyen et al., 2018). Memory-replay approaches include Experience Replay (Robins, 1995) and Deep Generative Replay (DGR) (Shin et al., 2017), as well as functional-regularization or distillation methods such as Functional Regularization for CL (FRCL) (Titsias et al., 2020), Functional Regularisation of Memorable Past (FROMP) (Pan et al., 2020), Incremental Classifier and Representation Learning (iCaRL) (Rebuffi et al., 2017), and Dark Experience Replay (DER) (Buzzega et al., 2020).

**Continual learning settings.** There are three distinct continual learning settings (Van de Ven & Tolias, 2019): 1) *task-incremental learning*, where the task identity is known, allowing us to have task-specific components such as a multiple output heads; 2) *domain-incremental learning*, where the task identity is not known and the model is not required to infer the task identity, which is typically solved by single-headed architectures, and 3) *class-incremental learning*, where the task identity is not known and is required to be inferred by the model, which includes common real-world problems such as sequentially learning new object classes (Van de Ven & Tolias, 2019). In this work, we exclusively focus on task-incremental learning, but it is also possible to extend our approach to other settings.

**Knowledge-adaptation priors (K-priors)** Khan & Swaroop (2021) is a general approach for general, quick and accurate model adaptation that unifies and generalizes many previous methods. In addition to our main technical contribution of improving K-priors by correcting for two major sources of error, we substantially expand upon Khan & Swaroop (2021) in two aspects: 1) They only consider one-step adaptation settings (i.e. with 2 tasks), and do not handle the accumulation of error as the number of tasks increases. It is especially important to assess more realistic CL scenarios with longer task sequences, which is exactly the focus of this work. 2) Their experiments are limited to small-scale models and datasets, the largest only involving a small CNN (1.2M parameters) on CIFAR-10 (10 classes, 60K data points). In contrast, we demonstrate scalability up to a ResNet-18 (11M parameters) on ImageNet (1000 classes, 1.2M data points).

## 3 Continual learning methods

We focus on a continual learning (CL) problem to incrementally learn from a sequence of data sets $\mathcal{D}_1, \dots, \mathcal{D}_T$, corresponding to a total of $T$ tasks. This is different from the commonly used batch-training in deep learning, where data from all the tasks is assumed to be available at all times during training. CL is challenging as the model needs to repeatedly adapt to new tasks, while not forgetting previously-gathered knowledge. Our goal is to get performance as close as possible to the model that is batch-trained on data from all tasks.

Formally, consider a supervised learning problem with $\mathcal{D}_t$ containing $N$ input-output pairs $(\boldsymbol{x}_i, y_i)$, with $\boldsymbol{x}_i \in \mathbb{R}^D$ and $y_i \in \mathcal{Y}$, and we wish to train a model with output $f_{\boldsymbol{w}}(\boldsymbol{x}_i)$ (also denoted as $f_{\boldsymbol{w}}^i$), and a $P$-length parameter $\boldsymbol{w}$ in a space $\mathcal{W} \subset \mathbb{R}^P$. Then, at any given task $t$, the best possible model parameters $\boldsymbol{w}_t$ can be obtained by training on all the data examples $\mathcal{D}_{1:t} = \cup_{i=1}^t \mathcal{D}_i$, for example, by solving the optimization problem shown below,

$$\boldsymbol{w}_t = \underset{\boldsymbol{w} \in \mathcal{W}}{\arg\min} \; \ell_t^{\text{batch}}(\boldsymbol{w}), \qquad \text{where } \ell_t^{\text{batch}}(\boldsymbol{w}) = \sum_{i \in \mathcal{D}_{1:t}} \ell\big(y_i, \sigma(f_{\boldsymbol{w}}^i)\big) + \mathcal{R}(\boldsymbol{w}). \tag{1}$$

We assume the loss $\ell(y_i, \sigma(f^i))$ is defined through the log-likelihood of an exponential family distribution (e.g., cross-entropy), with $\sigma(f)$ being a transformation of the model outputs (e.g., softmax) defined using the (inverse) link-function of the distribution. We denote the regularizer by $\mathcal{R}(\boldsymbol{w})$, and in what follows, we will use an $L_2$ regularizer $\mathcal{R}(\boldsymbol{w}) = \frac{1}{2}\delta\|\boldsymbol{w}\|^2$, with $\delta \geq 0$; other regularizers can also be used.

The main challenge in CL is to remember the useful past knowledge extracted from previous tasks $\mathcal{D}_{1:t-1}$, and reuse it while training over the new task $\mathcal{D}_t$ to get as close as possible to $\boldsymbol{w}_t$. We would like a compact summary of the past knowledge, because we are not allowed to store all past data. We will now describe two strategies used in the literature, based on weight-regularization and experience replay, and discuss the challenges in combining them.

Weight-regularization approaches aim to keep the new weight-vector close to the old one, hoping that this will help avoid forgetting and facilitate knowledge reuse. The most common is Elastic-Weight Consolidation (EWC) (Kirkpatrick et al., 2017), which minimizes the following objective using the previous weight $\boldsymbol{w}_{t-1}$,

$$\ell_t^{\text{weight}}(\boldsymbol{w}) = \sum_{i \in \mathcal{D}_t} \ell\big(y_i, \sigma(f_{\boldsymbol{w}}^i)\big) + \frac{\lambda}{2}(\boldsymbol{w} - \boldsymbol{w}_{t-1})^\top \mathbf{F}_{t-1}(\boldsymbol{w} - \boldsymbol{w}_{t-1}), \tag{2}$$

where the second term is a quadratic regularizer with a weight-importance matrix $\mathbf{F}_{t-1}$, and $\lambda \geq 0$ is a trade-off hyperparameter. The simplest choice of $\mathbf{F}_{t-1}$ is to use the diagonal of a generalized Gauss-Newton (GGN) matrix (Martens, 2020). The GGN (over previous tasks) is defined as,

$$\mathbf{G}_{t-1}(\mathcal{D}_{1:t-1}) := \sum_{i \in \mathcal{D}_{1:t-1}} \left[\nabla f_{\boldsymbol{w}_{t-1}}^i\right] \sigma'(f_{\boldsymbol{w}_{t-1}}^i) \left[\nabla f_{\boldsymbol{w}_{t-1}}^i\right]^\top, \tag{3}$$

where $\nabla f_{\boldsymbol{w}_{t-1}}^i$ denotes the Jacobian, that is, derivative of $f_{\boldsymbol{w}}(\boldsymbol{x}_i)$ with respect to $\boldsymbol{w}$ at $\boldsymbol{w} = \boldsymbol{w}_{t-1}$, and $\sigma'(f^i)$ denotes the derivative of $\sigma(f)$ with respect to $f$ at $f = f^i$. Often, the regularizer $\delta$ is also added to the GGN matrix to reduce ill-conditioning. Computation of $\mathbf{F}_{t-1}$ can be done in an online fashion as the training proceeds over tasks (Schwarz et al., 2018). This method uses a compact representation of the past knowledge as we need to only store $\boldsymbol{w}_{t-1}$ and the diagonal matrix $\mathbf{F}_{t-1}$, requiring $O(P)$ memory. The method is also simple to implement within deep-learning codebases, requiring relatively little overhead. Other choices are possible for the importance matrix (Zenke et al., 2017; Aljundi et al., 2018; Benzing, 2022).

An alternative to weight-regularization methods is based on experience replay (Robins, 1995; Shin et al., 2017). Such methods store a subset of past data in a memory $\mathcal{M}_t$ and add it to the new data during training:

$$\ell_t^{\text{er}}(\boldsymbol{w}) = \sum_{i \in \mathcal{D}_t \cup \mathcal{M}_{t-1}} \ell\big(y_i, \sigma(f_{\boldsymbol{w}}^i)\big) + \mathcal{R}(\boldsymbol{w}).$$

This method can be accurate when the memory represents previous tasks' data well, because as $\mathcal{M}_{t-1} \to \mathcal{D}_{1:t-1}$, $\arg\min \ell_t^{\text{er}} \to \boldsymbol{w}_t$, but this often requires a large memory that grows with the number of tasks.

A popular approach to combine these two previous approaches is to use functional regularization (Titsias et al., 2020; Pan et al., 2020) where, instead of regularizing the weights, we regularize the function outputs at a few past input locations stored in a memory:

$$\ell_t^{\text{func}}(\boldsymbol{w}) = \sum_{i \in \mathcal{D}_t} \ell\big(y_i, \sigma(f_{\boldsymbol{w}}^i)\big) + \frac{\lambda}{2} \sum_{j \in \mathcal{M}_{t-1}} \ell\left(\sigma(f_{\boldsymbol{w}_{t-1}}^j), \sigma(f_{\boldsymbol{w}}^j)\right). \tag{4}$$

Some methods implement this via knowledge-distillation (Rebuffi et al., 2017; Buzzega et al., 2020). Some simply use the original loss (as written in Eq. (4)) while others use the squared loss (Benjamin et al., 2019). An advantage of functional regularization is that it does not require the labels associated with the inputs in $\mathcal{M}_{t-1}$, which enables the use of an arbitrary input $\boldsymbol{x}$ which is not restricted to be from $\mathcal{D}_{1:t-1}$. For example, we can use a deep generative model to generate pseudo-inputs (Shin et al., 2017), or learn them as in sparse Gaussian processes (Titsias, 2009).

Overall, we see that all these strategies have their complementary strengths: weight-regularization is compact, experience replay can be more accurate, and functional regularization can use arbitrary memory inputs. Combining these approaches can strike a good balance between performance and memory size, but at present, little has been done to find principled ways to combine them. One could simply add them together, but there are many choices one needs to make. For example, how should we choose the importance matrix and the specific forms of the regularizers? A simple re-weighting is likely not the best choice. Our goal in this paper is to provide a theoretically-motivated approach to answer such questions.

# 4 A principle of adaptation: gradient reconstruction of the past

We will use the recently proposed principle of adaptation by Khan & Swaroop (2021) to combine and improve the CL strategies discussed in Section 3. The principle suggests to reconstruct the gradient of the past objective by using a combination of weight- and function-space regularizers. Specifically, at task $t$, we consider minimizing (for some $\tau > 0$)

$$\ell_t^{\text{K-prior}}(\boldsymbol{w}) = \sum_{i \in \mathcal{D}_t} \ell\big(y_i, \sigma(f_{\boldsymbol{w}}^i)\big) + \tau\,\mathcal{K}_t(\boldsymbol{w}; \boldsymbol{w}_{t-1}, \mathcal{M}_{t-1}), \tag{5}$$

where $\mathcal{K}_t(\boldsymbol{w}; \boldsymbol{w}_{t-1}, \mathcal{M}_{t-1})$ is a regularizer that combines a weight-space regularizer using $\boldsymbol{w}_{t-1}$ and a function-space regularizer over the memory $\mathcal{M}_{t-1}$. The regularizer is called the Knowledge-adaptation prior (K-prior), and is designed with the goal to minimize the gradient reconstruction error of the past training objective. We will use $\tau = 1$ unless noted otherwise.

Specifically, at task $t$, the past training objective is $\ell_{t-1}^{\text{batch}}(\boldsymbol{w})$, and we want to design the prior to minimize the magnitude of the gradient error for all $\boldsymbol{w}$ (more details are in Appendix A):

$$\boldsymbol{e}_t(\boldsymbol{w}) = \nabla \ell_t^{\text{batch}}(\boldsymbol{w}) - \nabla \ell_t^{\text{K-prior}}(\boldsymbol{w}) = \nabla \ell_{t-1}^{\text{batch}}(\boldsymbol{w}) - \nabla \mathcal{K}_t(\boldsymbol{w}; \boldsymbol{w}_{t-1}, \mathcal{M}_{t-1}). \tag{6}$$

The loss $\ell_{t-1}^{\text{batch}}$ depends on all the past data $\mathcal{D}_{1:t-1}$, and our goal is to reconstruct its gradient by using the weight vector $\boldsymbol{w}_{t-1}$ and a memory $\mathcal{M}_{t-1}$. Khan & Swaroop (2021) showed that many existing adaptive strategies in machine learning for one-step adaptation tasks can be recovered from this principle. For example, the following K-prior with $\mathcal{M}_{t-1} = \mathcal{D}_{1:t-1}$ gives zero error $\boldsymbol{e}_t(\boldsymbol{w})$ for generalized-linear models $f_{\boldsymbol{w}}^i = \phi_i^\top \boldsymbol{w}$ with feature vectors $\phi_i = \phi(\boldsymbol{x}_i)$:

$$\mathcal{K}_t(\boldsymbol{w}; \boldsymbol{w}_{t-1}, \mathcal{M}_{t-1}) = \sum_{i \in \mathcal{M}_{t-1}} \ell\Big(\sigma(f_{\boldsymbol{w}_{t-1}}^i), \sigma(f_{\boldsymbol{w}}^i)\Big) + \frac{\delta}{2}(\boldsymbol{w} - \boldsymbol{w}_{t-1})^\top(\boldsymbol{w} - \boldsymbol{w}_{t-1}), \tag{7}$$

where $\delta \geq 0$ is the $L_2$ regularization constant. The K-prior does not use the labels, just like functional-regularization discussed earlier, yet the gradient can be reconstructed by using the predictions at the inputs locations $\boldsymbol{x}_i \in \mathcal{D}_{1:t-1}$. To see that $\boldsymbol{e}_t$ vanishes, consider the following derivation from Khan & Swaroop (2021, Eq. 10):

$$\begin{aligned}
\boldsymbol{e}_t(\boldsymbol{w}) &= \nabla \ell_{t-1}^{\text{batch}}(\boldsymbol{w}) - \left[ \sum_{i \in \mathcal{D}_{1:t-1}} \phi_i \left[ \sigma(f_{\boldsymbol{w}}^i) - \sigma(f_{\boldsymbol{w}_{t-1}}^i) \right] + \delta(\boldsymbol{w} - \boldsymbol{w}_{t-1}) \right] \\
&= \cancel{\nabla \ell_{t-1}^{\text{batch}}(\boldsymbol{w})} - \underbrace{\left[ \sum_{i \in \mathcal{D}_{1:t-1}} \phi_i \left[ \sigma(f_{\boldsymbol{w}}^i) - y_i) \right] + \delta \boldsymbol{w} \right]}_{= \nabla \ell_{t-1}^{\text{batch}}(\boldsymbol{w}).} - \underbrace{\left[ \sum_{i \in \mathcal{D}_{1:t-1}} \phi_i \left[ \cancel{\sigma(f_{\boldsymbol{w}_{t-1}}^i)} - y_i \right] + \delta(\boldsymbol{w}_{t-1}) \right]}_{= \nabla \ell_{t-1}^{\text{batch}}(\boldsymbol{w}_{t-1}) = 0.} \\
&= 0,
\end{aligned}$$

where the second line is obtained by using Eq. (16) and noting that for a GLM, $\nabla f_{\boldsymbol{w}}^i = \phi_i$, and the third line is obtained by adding and subtracting outputs $y_i$ in the first term of $\nabla \mathcal{K}_t$. In the third line, the second term is equal to 0 because $\boldsymbol{w}_{t-1}$ is a minimizer of $\ell_{t-1}^{\text{batch}}$ and therefore $\nabla \ell_{t-1}^{\text{batch}}(\boldsymbol{w}_{t-1}) = 0$. Many other similar results are shown by Khan & Swaroop (2021) for other models, such as support vector machines, Gaussian processes, knowledge distillation, functional-regularization, and the memory-based methods used in continual learning.

K-prior is promising, but there are still multiple issues with the work of Khan & Swaroop (2021), which we address in this paper. First, they did not consider the sequential setup such as continual learning. Second, the error incurred by their K-prior, of form Eq. (7), is non-zero for neural networks: see Khan & Swaroop (2021, Sec. 4.2). For the continual learning problem, this can be disastrous because errors can accumulate quickly over tasks, deteriorating performance. Third, the weight-regularizer in Eq. (7) ignores the weight

importance, which is commonly used in other works (Kirkpatrick et al., 2017; Schwarz et al., 2018; Ritter et al., 2018) and can improve performance.

In this paper, we will address these issues, combining and improving regularization and memory-based methods. We will start with functional-regularization, and then add a weight regularizer and experience replay to decrease its error. This will yield a prior with lower error than each individual method on its own.

# 5 A new improved K-prior

Using the principle described in the previous section, we will now design a prior that combines a weight regularizer, a functional regularizer, and experience replay (Fig. 1). As described in the previous section, our goal is to minimize the gradient reconstruction error of the past training objective, i.e., the difference in the gradient of the loss for the full batch method and the K-priors loss with limited memory, see Eq. (6). The functional regularizer is based on knowledge distillation, and combined with an EWC-style quadratic regularizer which uses a specific importance vector to minimize the error in the K-prior of Eq. (7). An additional experience-replay term further reduces the error by storing the labels for a subset of the memory set. We will see that each piece contributes to the reduction of a different type of error, and the combination overall gives lower gradient reconstruction error than each individual component.

## 5.1 The error in the K-prior Eq. (7) when using a limited memory

We start by analyzing the gradient error in the K-prior of Eq. (7) when it is defined with a limited memory $\mathcal{M}_{t-1}$, instead of the full data $\mathcal{D}_{1:t-1}$. As shown in Appendix A, the error is given as follows,

$$
\boldsymbol{e}_t(\boldsymbol{w}) = \underbrace{\sum_{i \in \mathcal{D}_{1:t-1} \backslash \mathcal{M}_{t-1}} \nabla f^i_{\boldsymbol{w}} \left[ \sigma(f^i_{\boldsymbol{w}}) - \sigma(f^i_{\boldsymbol{w}_{t-1}}) \right]}_{:= \boldsymbol{e}^{\mathrm{mem}}_t(\boldsymbol{w}; \boldsymbol{w}_{t-1}, \mathcal{D}_{1:t-1} \backslash \mathcal{M}_{t-1})} + \underbrace{\sum_{i \in \mathcal{D}_{1:t-1}} \nabla f^i_{\boldsymbol{w}} r^i_{\boldsymbol{w}_{t-1}} + \delta \boldsymbol{w}_{t-1}}_{:= \boldsymbol{e}^{\mathrm{NN}}_t(\boldsymbol{w}; \boldsymbol{w}_{t-1}, \mathcal{D}_{1:t-1})}, \tag{8}
$$

where $r^i_{\boldsymbol{w}_t} = \sigma(f^i_{\boldsymbol{w}_t}) - y_i$ is the residual.[1] The first error term $\boldsymbol{e}^{\mathrm{mem}}_t$ arises due to the use of limited memory $\mathcal{M}_{t-1}$, and can be reduced to zero by increasing the memory size to include all past input examples. In contrast, the second error term $\boldsymbol{e}^{\mathrm{NN}}_t$ arises due to the use of neural networks, and reduces only when the network gets better in predicting the past data $\mathcal{D}_{1:t-1}$, that is, when the residuals go to zero. We will now show that $\boldsymbol{e}^{\mathrm{mem}}_t$ can be reduced by adding an EWC-style weight regularizer (Section 5.2), while $\boldsymbol{e}^{\mathrm{NN}}_t$ can be reduced by adding an experience replay term with a specific memory (Section 5.3).

## 5.2 Reducing $\mathrm{e}^{\mathrm{mem}}_t$ using an EWC-style regularizer

The error $\boldsymbol{e}^{\mathrm{mem}}_t$ can be reduced by using a first-order Taylor approximation of $\sigma(f^i_w)$ at $\boldsymbol{w}_{t-1}$,

$$
\sigma(f^i_{\boldsymbol{w}}) \approx \sigma(f^i_{\boldsymbol{w}_{t-1}}) + \sigma'(f^i_{\boldsymbol{w}_{t-1}})(\nabla f^i_{\boldsymbol{w}_{t-1}})^\top (\boldsymbol{w} - \boldsymbol{w}_{t-1}). \tag{9}
$$

Plugging this into the definition of the error, we get

$$
\boldsymbol{e}^{\mathrm{mem}}_t \approx \underbrace{\left[ \sum_{i \in \mathcal{D}_{1:t-1} \backslash \mathcal{M}_{t-1}} \left[ \nabla f^i_{\boldsymbol{w}_{t-1}} \right] \sigma'(f^i_{\boldsymbol{w}_{t-1}}) \left[ \nabla f^i_{\boldsymbol{w}_{t-1}} \right]^\top \right]}_{= \mathbf{G}_{t-1}(\mathcal{D}_{1:t-1} \backslash \mathcal{M}_{t-1})} (\boldsymbol{w} - \boldsymbol{w}_{t-1}),
$$

where we use the definition of the GGN matrix given in Eq. (3), and approximate $\nabla f^i_{\boldsymbol{w}_{t-1}} \approx \nabla f^i_{\boldsymbol{w}}$. The right hand side is equal to the gradient of an EWC-style regularizer,

$$
c^{\mathrm{mem}}_t = \tfrac{1}{2}(\boldsymbol{w} - \boldsymbol{w}_{t-1})^\top \mathbf{G}_{t-1}(\mathcal{D}_{1:t-1} \backslash \mathcal{M}_{t-1})(\boldsymbol{w} - \boldsymbol{w}_{t-1}), \tag{10}
$$

---

[1] Note that the combined error term in Eq. (8) unifies the limited memory error term and the neural network error term stated in, respectively, Eq. (11) and Eq. (16) in Khan & Swaroop (2021).

which uses the GGN over the past data but, importantly, excludes the memory $\mathcal{M}_{t-1}$ from it.[2] We can now define a new K-prior by adding the correction term to Eq. (5), giving,

$$\mathcal{K}_t^{\text{mem}}(\boldsymbol{w}; \boldsymbol{w}_{t-1}, \mathbf{F}_{t-1}, \mathcal{M}_{t-1}) = \sum_{i \in \mathcal{M}_{t-1}} \ell\Big(\sigma(f_{\boldsymbol{w}_{t-1}}^i), \sigma(f_{\boldsymbol{w}}^i)\Big) + \tfrac{1}{2}(\boldsymbol{w} - \boldsymbol{w}_{t-1})^\top \mathbf{F}_{t-1}(\boldsymbol{w} - \boldsymbol{w}_{t-1}), \qquad (11)$$

where we define $\mathbf{F}_{t-1} = \mathbf{G}_{t-1}(\mathcal{D}_{1:t-1} \backslash \mathcal{M}_{t-1}) + \delta \mathbf{I}$. By correcting for the error term, this K-prior reduces the error introduced in the K-prior of Eq. (7) due to a limited memory. Similarly to Online EWC (Schwarz et al., 2018), we can use a diagonal approximation to the GGN, and update it online.

The new K-prior not only reduces the gradient error, but also is more general because other regularizers are obtained as special cases by changing the memory. For an empty memory $\mathcal{M}_{t-1} = \emptyset$, it reduces to the EWC regularizer of Eq. (2): the first term in Eq. (11) disappears and the importance $\mathbf{F}_{t-1}$ is the GGN defined over all the past data, plus $\delta \mathbf{I}$. On the other hand, when the memory includes all past data, it reduces to the original K-prior in Eq. (7). When using limited memory, the new K-prior combines the functional and weight regularizers in a way to reduce the error in both EWC and the original K-prior.

### 5.3 Reducing $\mathrm{e}_t^{\mathrm{NN}}$ using experience replay

The $\boldsymbol{e}_t^{\mathrm{NN}}$ term depends on the mistakes made on the past data, and can be corrected by including an additional memory, denoted by $\mathcal{M}_{2,t-1}$, of the past data where mistakes are significant. We first note that the error is equivalent to the gradient of the following,

$$\boldsymbol{c}_t^{\mathrm{NN}} := \sum_{i \in \mathcal{M}_{2,t-1}} f_{\boldsymbol{w}}^i r_{\boldsymbol{w}_{t-1}}^i + \tfrac{1}{2}\delta \boldsymbol{w}^\top \boldsymbol{w}_{t-1}, \qquad (12)$$

when $\mathcal{M}_{2,t-1}$ is set to all the past data. Therefore, by adding $\mathcal{K}_t^{\mathrm{NN}} = \mathcal{K}_t + \boldsymbol{c}_t^{\mathrm{NN}}$ with a subset of past data $\mathcal{M}_{2,t-1}$, we reduce the error simply to $\sum_{i \in \mathcal{D}_{1:t-1} \backslash \mathcal{M}_{2,t-1}} \left[ \nabla f_{\boldsymbol{w}}^i r_{\boldsymbol{w}_{t-1}}^i \right]$.

### 5.4 K-prior with EWC-style regularizer and experience replay

Our new improved K-prior is obtained by simply correcting the K-prior from Eq. (7) by adding the correction terms from Eq. (10) and Eq. (12), that is,

$$\mathcal{K}_t^{\text{new}} = \mathcal{K}_t + \boldsymbol{c}_t^{\mathrm{NN}} + \boldsymbol{c}_t^{\text{mem}} = \sum_{i \in \mathcal{M}_{t-1}} \ell\Big(\sigma(f_{\boldsymbol{w}_{t-1}}^i), \sigma(f_{\boldsymbol{w}}^i)\Big) + \sum_{i \in \mathcal{M}_{2,t-1}} f_{\boldsymbol{w}}^i r_{\boldsymbol{w}_{t-1}}^i$$
$$+ \tfrac{1}{2}\delta \boldsymbol{w}^\top \boldsymbol{w}_{t-1} + \tfrac{1}{2}(\boldsymbol{w} - \boldsymbol{w}_{t-1})^\top \mathbf{F}_{t-1}(\boldsymbol{w} - \boldsymbol{w}_{t-1}). \qquad (13)$$

We make a specific choice of the memory: we choose $\mathcal{M}_{2,t-1}$ to be a subset of $\mathcal{M}_{t-1}$. This, as we show now, simplifies the computation and brings experience replay into our new K-prior.

Consider the first two terms from our improved K-prior in Eq. (13),

$$\sum_{i \in \mathcal{M}_{t-1}} \ell\Big(\sigma(f_{\boldsymbol{w}_{t-1}}^i), \sigma(f_{\boldsymbol{w}}^i)\Big) + \sum_{i \in \mathcal{M}_{2,t-1}} f_{\boldsymbol{w}}^i r_{\boldsymbol{w}_{t-1}}^i.$$

---

[2]Note that the derivation of Eq. (10) follows the derivation of Eq. (12) in Khan & Swaroop (2021).

The gradient of these two terms can be simplified when the second memory is a subset of the first one,

$$
\nabla \left[ \sum_{i \in \mathcal{M}_{t-1}} \ell\left( \sigma(f^i_{\boldsymbol{w}_{t-1}}), \sigma(f^i_{\boldsymbol{w}}) \right) + \sum_{i \in \mathcal{M}_{2,t-1}} f^i_{\boldsymbol{w}} r^i_{\boldsymbol{w}_{t-1}} \right]
$$

$$
= \sum_{i \in \mathcal{M}_{t-1}} \nabla f^i_{\boldsymbol{w}} \left[ \sigma(f^i_{\boldsymbol{w}}) - \sigma(f^i_{\boldsymbol{w}_{t-1}}) \right] + \sum_{i \in \mathcal{M}_{2,t-1}} \nabla f^i_{\boldsymbol{w}} r^i_{\boldsymbol{w}_{t-1}}
$$

$$
= \sum_{i \in \mathcal{M}_{t-1} \setminus \mathcal{M}_{2,t-1}} \nabla f^i_{\boldsymbol{w}} \left[ \sigma(f^i_{\boldsymbol{w}}) - \sigma(f^i_{\boldsymbol{w}_{t-1}}) \right] + \sum_{i \in \mathcal{M}_{2,t-1}} \nabla f^i_{\boldsymbol{w}} \left[ \sigma(f^i_{\boldsymbol{w}}) - \sigma(f^i_{\boldsymbol{w}_{t-1}}) \right] + \nabla f^i_{\boldsymbol{w}} r^i_{\boldsymbol{w}_{t-1}}
$$

$$
= \sum_{i \in \mathcal{M}_{t-1} \setminus \mathcal{M}_{2,t-1}} \nabla f^i_{\boldsymbol{w}} \left[ \sigma(f^i_{\boldsymbol{w}}) - \sigma(f^i_{\boldsymbol{w}_{t-1}}) \right] + \sum_{i \in \mathcal{M}_{2,t-1}} \nabla f^i_{\boldsymbol{w}} \left[ \sigma(f^i_{\boldsymbol{w}}) - y_i \right]
$$

where in the second line we used the expression in Eq. (16) for the gradient of the (exponential-family) loss, and where in the last line we used the definition of the residual $r^i_{\boldsymbol{w}_{t-1}} = \sigma(f^i_{\boldsymbol{w}_{t-1}}) - y_i$ to simplify. This gradient is equal to the gradient of a sum of a functional-regularizer term and an experience replay term,

$$
\sum_{i \in \mathcal{M}_{t-1} \setminus \mathcal{M}_{2,t-1}} \ell\left( \sigma(f^i_{\boldsymbol{w}_{t-1}}), \sigma(f^i_{\boldsymbol{w}}) \right) + \sum_{i \in \mathcal{M}_{2,t-1}} \ell\left( y_i, \sigma(f^i_{\boldsymbol{w}}) \right), \tag{14}
$$

again using the expression for the loss gradient in Eq. (16). Therefore, we can rewrite the new K-prior as the following combination,

$$
\mathcal{K}^{\mathrm{new}}_t(\boldsymbol{w}; \boldsymbol{w}_{t-1}, \mathbf{F}_{t-1}, \mathcal{M}_{t-1}) = \underbrace{\sum_{i \in \mathcal{M}_{1,t-1}} \ell\left( \sigma(f^i_{\boldsymbol{w}_{t-1}}), \sigma(f^i_{\boldsymbol{w}}) \right)}_{\text{Functional-regularization}} + \underbrace{\sum_{i \in \mathcal{M}_{2,t-1}} \ell\left( y_i, \sigma(f^i_{\boldsymbol{w}}) \right)}_{\text{Experience replay}}
$$
$$
+ \underbrace{\tfrac{1}{2}(\boldsymbol{w} - \boldsymbol{w}_{t-1})^\top \mathbf{F}_{t-1} (\boldsymbol{w} - \boldsymbol{w}_{t-1}) + \tfrac{1}{2}\delta \boldsymbol{w}^\top \boldsymbol{w}_{t-1}}_{\text{Weight-regularization}}, \tag{15}
$$

where $\mathcal{M}_{1,t-1} = \mathcal{M}_{t-1} \setminus \mathcal{M}_{2,t-1}$ and $\mathcal{M}_{2,t-1}$ are two disjoint subsets of the memory, and where we define $\mathbf{F}_{t-1} = \mathbf{G}_{t-1}(\mathcal{D}_{1:t-1} \setminus \mathcal{M}_{t-1}) + \delta \mathbf{I}$. The new prior combines a weight regularizer, a functional regularizer, and experience replay. The functional regularizer is based on knowledge distillation, and combined with an EWC-style quadratic regularizer which uses an importance vector obtained over all the past data but excluding the memory $\mathcal{M}_{t-1} = \mathcal{M}_{1,t-1} \cup \mathcal{M}_{2,t-1}$. Using this combination, the new prior gives lower gradient reconstruction error than each method alone. We will refer to our method as `EWC+FR+Replay` (Elastic Weight Consolidation + Function Regularization + Experience Replay).

In our experiments, we select the memories as follows. We first randomly (uniformly) sample the memory set $\mathcal{M}_{t-1}$ from all data $\mathcal{D}_{1:t-1}$, and then randomly split it into the two memory subsets $\mathcal{M}_{1,t-1}$ and $\mathcal{M}_{2,t-1}$; for simplicity, we use equally sized subsets, i.e., $|\mathcal{M}_{1,t-1}| = |\mathcal{M}_{2,t-1}|$. We found this simple selection strategy to work surprisingly well, and leave the study of more sophisticated approaches (as well as of settings where $|\mathcal{M}_{1,t-1}| \neq |\mathcal{M}_{2,t-1}|$) for future work.

## 6 Empirical evaluation

We first describe the general experimental setup used throughout our evaluation. We then present empirical results showing the practical benefits of our proposed `EWC+FR+Replay` method. We focus on multi-class classification in the task-incremental learning setting. In particular, we evaluate on three continual learning benchmarks with increasing size and thus difficulty: 1) Split-CIFAR (medium-scale), 2) Split-TinyImageNet (medium-to-large-scale), and 3) ImageNet-1000 (large-scale). See Appendix B for more details on the experiments (e.g. hyperparameters used).

### 6.1 Experimental setup

**Continual learning setup.** We mostly follow previous works on CL. We consider the common multi-head task-incremental (Van de Ven & Tolias, 2019) setting with known task identities: each method is trained

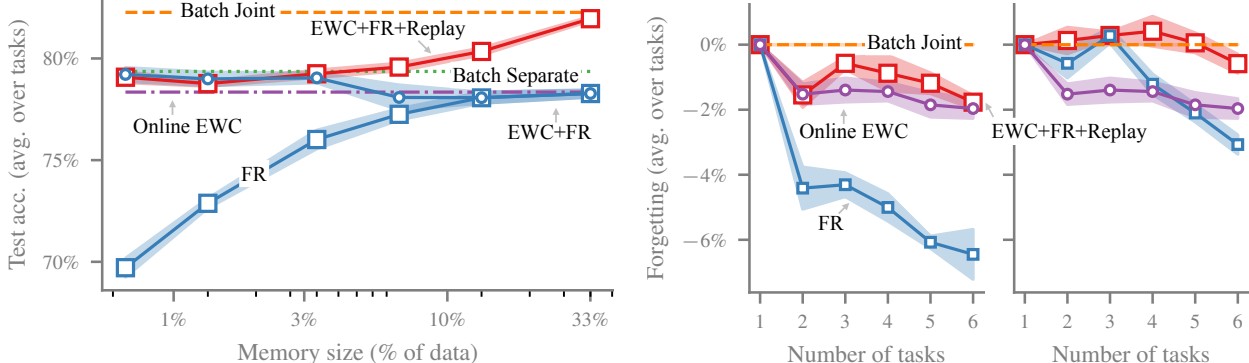

Figure 2: **Results on Split-CIFAR.** `EWC+FR+Replay` is superior across memory sizes, closely approaching `Batch Joint` for memory size 33.3% (left; x-axis log-scaled). It also suffers less from FORGETTING (relative to `Batch Joint`) with a growing number of tasks, for memory sizes 0.7% (middle) and 33.3% (right).

sequentially on all tasks with a separate classification head per task, and is told which task an input belongs to both at train and test time. We report test accuracy of the final model trained on the entire task sequence (averaged over the test sets of all observed tasks). We also compute average FORGETTING (aka backward-transfer) as defined in Lopez-Paz & Ranzato (2017), which captures the (average) difference in accuracy between when a task is first trained and after the final task. We plot mean $\pm$ standard error over three seeds, and assess performance across a wide range of memory sizes.

**Methods.** In addition to our proposed `EWC+FR+Replay` method, we evaluate five relevant baselines for comparison. In summary, we consider the following methods:

1. `EWC+FR+Replay`. Our proposed regularizer which combines the K-prior function regularizer over $\mathcal{M}_1$ with the EWC-style weight regularizer and the experience replay term over $\mathcal{M}_2$.

2. `FR`. The original K-prior function regularizer over $\mathcal{M}_1$ as proposed by Khan & Swaroop (2021), *without* the EWC-style weight regularization term and *without* the experience replay term over $\mathcal{M}_2$.

3. `EWC+FR`. A regularizer combining the K-prior function regularizer over $\mathcal{M}_1$ with *only* the EWC-style weight regularization, i.e. *without* the experience replay term over $\mathcal{M}_2$.

4. `Batch Joint`. Joint batch training of a single multi-head model across the data of all tasks, i.e. the optimal CL solution which serves as an upper bound we wish to approach.

5. `Batch Separate`. Independent batch training of a *separate* model for each task.

6. `Online EWC` (Schwarz et al., 2018), which has the same weight regularizer as `EWC+FR+Replay`, but *without* the function regularizer over $\mathcal{M}_1$ or the experience replay term over $\mathcal{M}_2$.

## 6.2 Results on Split-CIFAR

**Setup.** Split-CIFAR (Zenke et al., 2017) has 6 tasks with 10 classes each. The first task is CIFAR-10 (Krizhevsky et al., 2009) with 50,000 training and 10,000 test data points across 10 classes. The subsequent 5 tasks are taken sequentially from CIFAR-100 (Krizhevsky et al., 2009), each with 5,000 training and 1,000 test data points across 10 classes. In total, we thus have 90,000 data points. We use the same CifarNet model as Zenke et al. (2017); Pan et al. (2020): a multi-head CNN with 4 convolutional layers, followed by 2 dense layers with dropout, with ~1.2M model parameters in total. On each task, we train for 80 epochs using Adam with learning rate $10^{-3}$ and batch size 256.

**Results.** Fig. 2 shows our results on Split-CIFAR. We consider memory sizes between 100 and 5,000 per task; at 5,000, we thus store 10% of the data for task 1, and all data for tasks 2-5 (which in total thus

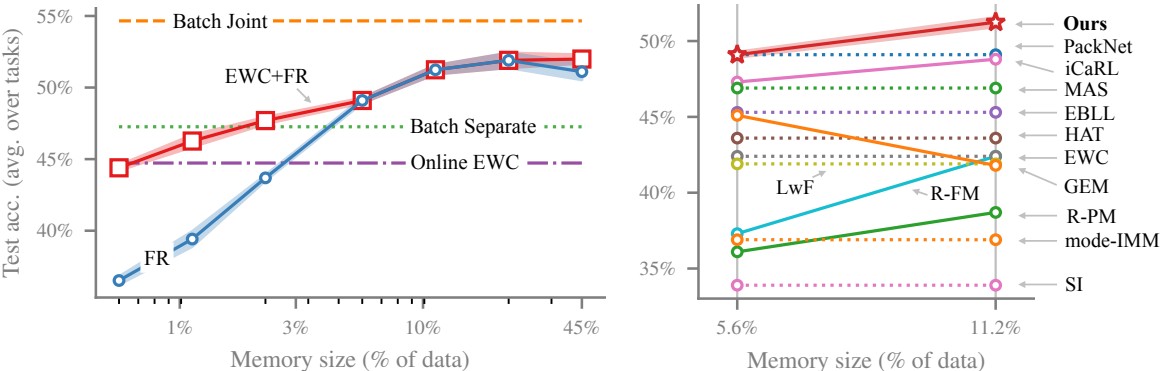

Figure 3: **Results on Split-TinyImageNet.** `EWC+FR` performs well across memory sizes (left; x-axis log-scaled) and compared to further strong baselines from Delange et al. (2021) (right).

corresponds to a third of all data). We see that `FR` performs poorly at small memory sizes and is much worse than `Online EWC`. While performance improves noticeably with growing memory size, `FR` remains far below `Batch Joint` even at memory size 33.3%. This confirms that when using a NN instead of a GLM, the theory behind `FR` (see Section 4) indeed ceases to hold. Adding the EWC-style weight regularizer (`EWC+FR`) substantially improves performance at small memory sizes, empirically confirming the property that `EWC+FR` converges to `Online EWC` for small memories (see Section 5.2). However, we also confirm that for large memories, `EWC+FR` converges to `FR`, resulting in a performance drop.

Finally, when we also add the experience replay term to obtain `EWC+FR+Replay`, performance is substantially boosted also at large memory sizes, actually enabling us to reach `Batch Joint` performance for memory size 33.3%. We see that `EWC+FR+Replay` obtains ∼97% of the batch performance with just ∼10% of the data, but does not degrade as memory size decreases (unlike `FR`). `EWC+FR+Replay` thus combines the complementary benefits of both error correction terms, i.e. of both EWC-style weight-regularization and experience replay, to significantly improve upon `FR` in both the small *and* large memory regime. Fig. 2 (mid & right) further shows that our method can leverage prior knowledge more effectively than other methods, thereby suffering less from FORGETTING with a growing number of tasks. This confirms that the two error correction terms in `EWC+FR+Replay` are particularly important for mitigating error accumulation across longer task sequences.

### 6.3 Results on Split-TinyImageNet

**Setup.** Following Delange et al. (2021), we construct Split-TinyImageNet by dividing TinyImageNet (Le & Yang, 2015) into a sequence of 10 tasks with 20 classes each (using the same random division as in Delange et al. (2021)). Each class has 500 data points split into training (80%) and validation (20%), and 50 test points (totalling to 110,000 points). We use the VGG-like BASE model from Delange et al. (2021) with 6 convolutional layers, 4 max pooling layers, and 2 dense layers, with a total of ∼3.5M parameters. On each task, we train for 70 epochs (with early stopping and exponential learning rate decay, without regularization) using SGD with momentum 0.9 and batch size 200. This replicates the setup of Delange et al. (2021) to make our results directly comparable.[3]

**Results.** Fig. 3 shows our results on Split-TinyImageNet. We found that the experience replay error correction term does not help on this benchmark, so we representatively plot just `FR` and `EWC+FR`.[4] We again see that `EWC+FR` can substantially improve over `FR` (especially at small memory sizes) and `Online EWC`, achieving ∼90% of the batch performance with ∼10% of the data (Fig. 3 left). It also compares favourably against a diverse range of other strong CL methods across all three CL paradigms: 1) memory/rehearsal – iCaRL (Rebuffi et al., 2017), GEM (Lopez-Paz & Ranzato, 2017), R-FM & R-PM (Delange et al., 2021),

---

[3]The only difference lies in the hyperparameter tuning procedure: while Delange et al. (2021) use their proposed online tuning algorithm, we resort to a standard grid search for simplicity; see Appendix B for details.

[4]This is likely because almost perfect train accuracy is attained on all tasks (see e.g. Table 14 in Delange et al. (2021)). Thus, $e_t^{NN}$ in Eq. (8) is close to zero, such that NN error correction cannot boost performance.

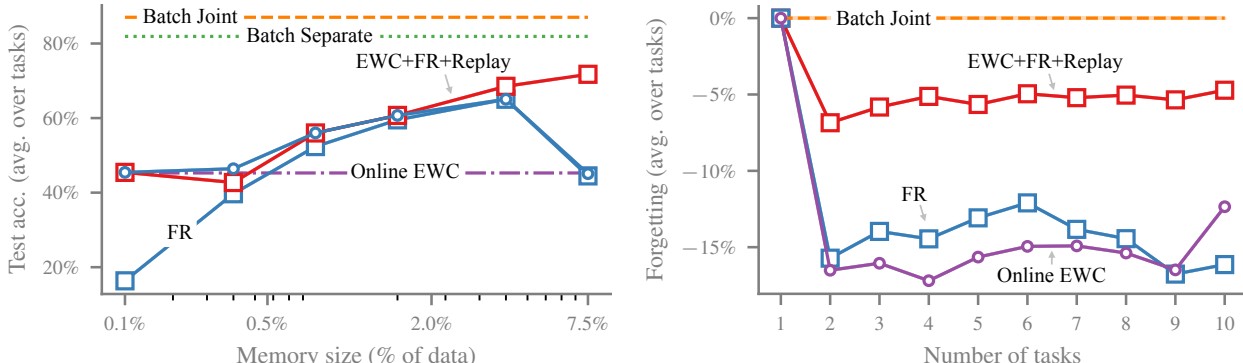

Figure 4: **Results on ImageNet-1000.** `EWC+FR+Replay` performs favorably across a range of memory sizes (left; x-axis log-scaled), and suffers less from FORGETTING (relative to `Batch Joint`) with an increasing number of tasks, here exemplary shown at the largest memory size of 7.5% (right).

2) weight-regularization – LwF (Li & Hoiem, 2017), EBLL (Rannen et al., 2017), EWC (Kirkpatrick et al., 2017), SI (Zenke et al., 2017), MAS (Aljundi et al., 2018), mode-IMM (Lee et al., 2017), and 3) architectural – PackNet (Mallya & Lazebnik, 2018), HAT (Serra et al., 2018) (Fig. 3 right).[5]

### 6.4 Results on ImageNet-1000

**Setup.** We consider the ImageNet-1000 benchmark proposed by Rebuffi et al. (2017), which randomly splits the full ImageNet dataset (Deng et al., 2009) of ∼1.2M data points into a sequence of 10 tasks with 100 classes and ∼120K data points each. Following Rebuffi et al. (2017), we use a ResNet-18 with ∼11M model parameters. For training on each task, we use the ImageNet reference training pipeline (with 40 epoch configuration) of the FFCV library (Leclerc et al., 2022).[6]

**Results.** Fig. 4 shows our results on ImageNet-1000. We consider memory sizes between 200 and 10K per task, where the latter amounts to 7.5% of the entire data. The observed trends qualitatively match those from previous experiments. In particular, `FR` underperforms for small memory sizes, and while it improves with increasing memory, it peaks at a 3.8% memory and then even starts declining. We hypothesize that this is again due to accumulation of the NN error, which might become more severe with a larger memory as more data points can contribute to the error. `EWC+FR` again improves accuracy for small memories, but does not help for large memories. Finally, correcting for the NN error by additionally including the experience replay term (`EWC+FR+Replay`) substantially boosts performance also at the large 7.5% memory. `EWC+FR+Replay` thus combines the benefits of both error correction terms to perform well across all memory sizes, achieving > 80% of the batch performance with a memory of < 10% of the past data. It also again suffers less from FORGETTING along the task sequence, demonstrating that it better mitigates error accumulation.

## 7 Conclusion

In this work, we proposed to address the continual learning problem in a theoretically-grounded way by explicitly approximating the optimal model obtained via batch-training on all tasks jointly. To this end, we developed `EWC+FR+Replay`, a new continual learning method which efficiently re-uses prior knowledge to reconstruct the gradients of the past training objective as faithfully as possible. To achieve this, our method combines principles from function-regularization, weight-regularization, and experience replay to reduce the gradient-reconstruction error. Empirically, we demonstrated the effectiveness and scalability of `EWC+FR+Replay` across different memory sizes on common task-incremental continual learning benchmarks. In particular, we showed that our proposed `EWC+FR+Replay` approach can be less susceptible to catastrophic

---

[5]Results are from Delange et al. (2021); their total memory sizes [4500, 9000] correspond to [5.6%, 11.2%] of the data.

[6]For all details of the training procedure, see https://github.com/libffcv/ffcv-imagenet/.

forgetting and thus achieve better performance compared to various baselines. Notably, `EWC+FR+Replay` can achieve $> 80\%$ of the batch performance by utilizing a memory of $< 10\%$ of the past data on the task-incremental continual learning benchmarks considered. For future work, we aim to investigate how more sophisticated strategies for selecting the two memory sets can further boost performance of our method.

## Acknowledgements

We would like to thank Runa Eschenhagen for many helpful discussions throughout the project. ED acknowledges funding from the EPSRC and Qualcomm. MEK and RY are supported by the Bayes duality project, JST CREST Grant Number JPMJCR2112. JMHL acknowledges support from a Turing AI Fellowship under grant EP/V023756/1.

## Author Contributions Statement

List of Authors: Erik Daxberger (ED), Siddharth Swaroop (SS), Kazuki Osawa (KO), Rio Yokota (RY), Richard E. Turner (RET), José Miguel Hernández-Lobato (JMHL), Mohammad Emtiyaz Khan (MEK).

SS and MEK conceived the original idea. This was then discussed and further refined with ED, KO, RY, RET, JMHL. ED led the project. SS derived the combination of K-prior and EWC, and together with ED and MEK the method was extended to include replay. SS implemented the original K-prior algorithm. KO designed and implemented the distributed training pipeline. ED designed and conducted all the experiments, with continuous support from SS and KO and feedback from all other authors. MEK and RY provided the computational resources for running the experiments. ED wrote the first draft of the paper, which was then substantially revised by SS and MEK. RY, RET, JMHL gave regular feedback throughout the project.

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

## A   Derivation of error for K-priors with limited memory Eq. (8)

Recall that $f_{\boldsymbol{w}}^i = f_{\boldsymbol{w}}(\boldsymbol{x}_i)$ is a shorthand for the model outputs. For ease-of-notation, we will also use the shorthand $\sigma_{\boldsymbol{w}}^i = \sigma(f_{\boldsymbol{w}}^i) = \sigma(f_{\boldsymbol{w}}(\boldsymbol{x}_i))$ for the model predictions. Consider the following expression for the gradient of the (exponential-family) loss (Khan & Swaroop, 2021),

$$\nabla \ell(y_i, \sigma(f_{\boldsymbol{w}}^i)) = \nabla f_{\boldsymbol{w}}^i \left[ \sigma(f_{\boldsymbol{w}}^i) - y_i \right]. \tag{16}$$

We start by deriving the gradient error in Eq. (6),

$$
\begin{aligned}
&\nabla \ell_t^{\text{batch}}(\boldsymbol{w}) - \nabla \ell_t^{\text{K-prior}}(\boldsymbol{w}) \\
&= \nabla \left( \ell_t^{\text{batch}}(\boldsymbol{w}) - \ell_t^{\text{K-prior}}(\boldsymbol{w}) \right) \\
&\overset{(1),(5)}{=} \nabla \left( \sum_{i \in \mathcal{D}_t} \ell\left(y_i, \sigma_{\boldsymbol{w}}^i\right) + \ell_{t-1}^{\text{batch}}(\boldsymbol{w}) - \sum_{i \in \mathcal{D}_t} \ell\left(y_i, \sigma_{\boldsymbol{w}}^i\right) - \mathcal{K}(\boldsymbol{w}; \boldsymbol{w}_{t-1}, \mathcal{M}) \right) \\
&= \nabla \left( \ell_{t-1}^{\text{batch}}(\boldsymbol{w}) - \mathcal{K}(\boldsymbol{w}; \boldsymbol{w}_{t-1}, \mathcal{M}) \right) \\
&= \nabla \ell_{t-1}^{\text{batch}}(\boldsymbol{w}) - \nabla \mathcal{K}(\boldsymbol{w}; \boldsymbol{w}_{t-1}, \mathcal{M}).
\end{aligned}
$$

We then have,

$$
\begin{aligned}
&\nabla \left( \ell_{t-1}^{\text{batch}}(\boldsymbol{w}) - \mathcal{K}(\boldsymbol{w}; \boldsymbol{w}_{t-1}, \mathcal{M}) \right) \\
&\overset{(1),(7)}{=} \nabla \left( \sum_{i \in \mathcal{D}_{1:t-1}} \ell\left(y_i, \sigma_{\boldsymbol{w}}^i\right) - \sum_{i \in \mathcal{M}} \ell\left(\sigma_{\boldsymbol{w}_{t-1}}^i, \sigma_{\boldsymbol{w}}^i\right) + \tfrac{1}{2}\delta \|\boldsymbol{w}_{t-1}\|^2 \right) \\
&= \sum_{i \in \mathcal{D}_{1:t-1}} \nabla \ell\left(y_i, \sigma_{\boldsymbol{w}}^i\right) - \sum_{i \in \mathcal{M}} \nabla \ell\left(\sigma_{\boldsymbol{w}_{t-1}}^i, \sigma_{\boldsymbol{w}}^i\right) + \delta \boldsymbol{w}_{t-1} \\
&\overset{(16)}{=} \sum_{i \in \mathcal{D}_{1:t-1}} \nabla f_{\boldsymbol{w}}^i \left[ \sigma_{\boldsymbol{w}}^i - y_i \right] - \sum_{i \in \mathcal{M}} \nabla f_{\boldsymbol{w}}^i \left[ \sigma_{\boldsymbol{w}}^i - \sigma_{\boldsymbol{w}_{t-1}}^i \right] + \delta \boldsymbol{w}_{t-1} \\
&= \sum_{i \in \mathcal{D}_{1:t-1}} \nabla f_{\boldsymbol{w}}^i \left[ \sigma_{\boldsymbol{w}}^i - y_i + \sigma_{\boldsymbol{w}_{t-1}}^i - \sigma_{\boldsymbol{w}}^i + \sigma_{\boldsymbol{w}}^i - \sigma_{\boldsymbol{w}_{t-1}}^i \right] - \sum_{i \in \mathcal{M}} \nabla f_{\boldsymbol{w}}^i \left[ \sigma_{\boldsymbol{w}}^i - \sigma_{\boldsymbol{w}_{t-1}}^i \right] + \delta \boldsymbol{w}_{t-1} \\
&= \sum_{i \in \mathcal{D}_{1:t-1}} \nabla f_{\boldsymbol{w}}^i \left[ \sigma_{\boldsymbol{w}_{t-1}}^i - y_i \right] + \sum_{i \in \mathcal{D}_{1:t-1}} \nabla f_{\boldsymbol{w}}^i \left[ \sigma_{\boldsymbol{w}}^i - \sigma_{\boldsymbol{w}_{t-1}}^i \right] - \sum_{i \in \mathcal{M}} \nabla f_{\boldsymbol{w}}^i \left[ \sigma_{\boldsymbol{w}}^i - \sigma_{\boldsymbol{w}_{t-1}}^i \right] + \delta \boldsymbol{w}_{t-1} \\
&= \sum_{i \in \mathcal{D}_{1:t-1} \setminus \mathcal{M}} \nabla f_{\boldsymbol{w}}^i \left[ \sigma_{\boldsymbol{w}}^i - \sigma_{\boldsymbol{w}_{t-1}}^i \right] + \sum_{i \in \mathcal{D}_{1:t-1}} \nabla f_{\boldsymbol{w}}^i \left[ \sigma_{\boldsymbol{w}_{t-1}}^i - y_i \right] + \delta \boldsymbol{w}_{t-1} \\
&= \sum_{i \in \mathcal{D}_{1:t-1} \setminus \mathcal{M}} \nabla f_{\boldsymbol{w}}^i \left[ \sigma_{\boldsymbol{w}}^i - \sigma_{\boldsymbol{w}_{t-1}}^i \right] + \sum_{i \in \mathcal{D}_{1:t-1}} \nabla f_{\boldsymbol{w}}^i r_{\boldsymbol{w}_{t-1}}^i + \delta \boldsymbol{w}_{t-1}
\end{aligned}
$$

where $r_{\boldsymbol{w}_{t-1}}^i = \sigma_{\boldsymbol{w}_{t-1}}^i - y_i$ is the residual of the $i$'th input using the past model parameters $\boldsymbol{w}_{t-1}$.[7]

## B   Experiment details

For all methods in all experiments, we tuned hyperparameters in the common way by conducting a (exponentially-spaced) grid search and evaluating performance on a held-out validation set. In particular, we tune the following hyperparameters: a temperature parameter $T$ to scale the logits in the experience replay term (as commonly-done in knowledge distillation, see Khan & Swaroop (2021) for a discussion), a trade-off parameter $\tau$ in front of the K-prior-style function regularization term, and a trade-off parameter $\lambda$ in front of the EWC-style weight regularization term.

---

[7]Note that this derivation follows the derivation in Appendix C in Khan & Swaroop (2021).

We used the following tuned values for those hyperparameters: for Split-CIFAR, we have $T = 2.0$ for all methods, $\tau = 0.1$ for `FR`, $\tau = 0.25$ and $\lambda = 2.0$ for `EWC+FR` and `EWC+FR+Replay`, and $\lambda = 10.0$ for `Online EWC`. For Split-TinyImageNet, we have $T = 1.0$ and $\tau = 16.0$ for all methods. We found that a fixed $\lambda$ across tasks does not work well for this benchmark, so we tuned a separate $\lambda$ per task, resulting in the sequence $[330, 85, 45, 30, 20, 15, 10, 10]$ for all methods. In contrast to our grid search procedure, Delange et al. (2021) use a dedicated iterative hyperparameter tuning strategy that trades off plasticity vs. stability. For ImageNet, we used $T = 1.0$, $\lambda = 1.0$ and $\tau = 0.16$ for all methods.

