# OpenReview forum: "Improving Continual Learning by Accurate Gradient Reconstructions of the Past"
_TMLR — Accepted by TMLR_

### Review · Reviewer_GymF · 2023-08-06

**Summary Of Contributions:**

The paper proposes a method that combines three existing continual learning (CL) approaches: elastic weight consolidation (EWC), functional regularization and experience replay. The choice of these three approaches is motivated by the so-called principle of adaptation. This previously proposed principle adds a regularizing term (K-prior) to the training objective. The prior is chosen to minimize the magnitude of the gradient error. The authors demonstrate how the combination of EWC, experience replay and functional regularization can be used to reduce different parts of the gradient error. Empirical evaluation demonstrates that this combination of continual learning techniques improves the performance in task-incremental continual learning on various benchmarks.


**Audience:**

Yes

**Claims And Evidence:**

No

**Requested Changes:**

**I have one major concern that is critical for my final evaluation of the paper:**

The authors make the following claim in the introduction: “...we provide a principled approach to combine and improve the two strategies” [regularization and experience replay]. Further, at the end of section 3, the authors write: “...how should we choose the importance matrix, the memory set, and the specific forms of the regularizers? Our goal in this paper is to provide a principled approach to answer such questions”.

I do not agree that the paper fulfils these claims. The paper demonstrates that the EWC form of regularization reduces a certain part of the gradient error, while experience replay reduces another. This demonstrates the benefit of combining the two. However, the paper does not study other regularization-based CL methods (e.g.  Synaptic Intelligence or Memory Aware Synapses, etc.) from the same point of view. Various ways to select a replay buffer are not considered as well. That being said, I still have a question posed by the authors: how do we choose the importance matrix and the memory set?
 Furthermore, I would expect a principled approach to provide certain guidelines on how the two approaches could be weighted in the loss. For example, a larger memory buffer places more emphasis on the experience replay. Does one need to counteract that by reweighting the regularization part of the loss?

I would gladly hear the author's clarification on how these questions can be potentially answered using their framework. In the current state of the work, I do not think this can be called a principled approach and formulations in the paper should be softened.

**Minor concern**:
* In section 5.2 first-order Taylor approximation is used for the term $\sigma(f_{\omega}^i)$ and constant approximation is chosen for $\nabla f_{\omega}^i$. Why don’t we consider 1st order approximation for the whole $e^{mem}_t$ term? How would that influence the conclusions of the section?

**Recommended minor adjustments:**
* Introduction: Paragraphs 3 and 4 both start with the sentence that mentions “two strategies”. It should be known to the reader from the beginning of paragraph 2 that it refers to regularization and experience replay. However, I think, this cross-paragraph reference harms the “flow” of the paper.
* Equation (7) provides the definition of the gradient error. However, the derivation of the gradient error in Appendix A begins from a different expression. After a few transformations, we do arrive at the same thing as equation (7), but it is quite confusing at the beginning.
* Equation (17): the same letter $\delta$ is used in the definition of $F_{t-1}$ and in front of the $L_2$ regularization. Please, consider renaming one of them.

**Typos:**
* Page 3 (continual learning settings): “...task-incremental learning, where the task identity is known, allowing us **TO** have task-specific model”
* Page 3 (Knowledge-adaptation priors): “Khan & Swaroop (2021) **IS** a principled, foundational approach…”
* Equation (11): $\tfrac{1}{2}$ is missing


**Strengths And Weaknesses:**

Strengths:
* The authors propose to combine two continual learning techniques using theoretically grounded arguments.
* Most derivations are clear and easy to follow.

Weaknesses:
* I do not agree that in its current state, the paper describes a principled way to combine regularization and replay-based continual learning methods as claimed by the authors (see the requested change section).

---

> ### Author Response · Authors · 2023-08-30
> **Response to Reviewer GymF**
>
> > I do not agree that in its current state, the paper describes a principled way to combine regularization and replay-based continual learning methods as claimed by the authors. Requested change: formulations in the paper should be softened.
>
> We have softened the claims regarding our approach being “principled”; in particular, we made the following changes (please let us know in case we missed other occurrences):
> - Introduction, fourth paragraph: “we provide a principled approach” → “we provide a _theoretically-motivated_ approach”
> - Introduction, last paragraph: “This approach is principled” → “This approach is _theoretically-motivated_”
> - Section 3, last paragraph: “Our goal in this paper is to provide a principled approach” → “Our goal in this paper is to provide a _theoretically-motivated_ approach”
>
> > The paper demonstrates that the EWC form of regularization reduces a certain part of the gradient error, while experience replay reduces another. This demonstrates the benefit of combining the two. However, the paper does not study other regularization-based CL methods (e.g. Synaptic Intelligence or Memory Aware Synapses, etc.) from the same point of view. Various ways to select a replay buffer are not considered as well. That being said, I still have a question posed by the authors: how do we choose the importance matrix and the memory set?
>
> Regarding the importance matrix, we do suggest choosing the Hessian / Fisher, which follows naturally from the theory. In fact, both SI and MAS can be viewed as alternative approximations to the Hessian / Fisher, as shown in the paper titled “Unifying Regularisation Methods for Continual Learning” (https://arxiv.org/abs/2006.06357).
>
> We agree that we do not provide a solution to choosing the memory set, and have removed the respective claim: in particular, we made the following change:
> - Section 3, last paragraph: “how should we choose the importance matrix, _the memory set,_ and the specific forms of the regularizers” → “how should we choose the importance matrix and the specific forms of the regularizers”
>
> > I would expect a principled approach to provide certain guidelines on how the two approaches could be weighted in the loss. For example, a larger memory buffer places more emphasis on the experience replay. Does one need to counteract that by reweighting the regularization part of the loss?
>
> Note that our theoretical derivation suggests to use a weight of 1 for each term. Also, note that a larger memory buffer does not place more emphasis on the experience replay term, because the weight-regularization matrix is down-weighted by the right amount accordingly.
>
> Finally, we would like to thank you for providing the thorough list of minor suggestions, which we have incorporated in our revision.

---

### Review · Reviewer_ioCd · 2023-08-07

**Summary Of Contributions:**

This paper proposes to combine regularization with replay to improve continual learning. Numerical experiments demonstrate the improvement of the proposed method.

**Audience:**

Yes

**Claims And Evidence:**

Yes

**Requested Changes:**

The authors are encouraged to provide toy examples with theoretical justifications on why regularization is important to address their novelty of the paper.

**Strengths And Weaknesses:**

Strength: The paper is written clearly and easy to understand.

Weakness: The major concern is about the novelty of this paper. While replay is a specific method in continual learning, it is well known that regularization can improve the generalization for most machine learning algorithms. As a result, without more in-depth theoretical insights on why regularization particularly fits continual learning, the novelty of the proposed method is insufficient.

---

> ### Author Response · Authors · 2023-08-30
> **Response to Reviewer ioCd**
>
> > The major concern is about the novelty of this paper. While replay is a specific method in continual learning, it is well known that regularization can improve the generalization for most machine learning algorithms. As a result, without more in-depth theoretical insights on why regularization particularly fits continual learning, the novelty of the proposed method is insufficient. The authors are encouraged to provide toy examples with theoretical justifications on why regularization is important to address their novelty of the paper.
>
> We believe you are misunderstanding this point. While you are right that previous papers have suggested combining different CL methods in an ad-hoc way (which intuitively makes sense), no previous paper has provided a mathematical justification for combining the different approaches, to the best of our knowledge. Therefore, the novelty of our paper lies in providing a theoretical derivation for how to combine the different methods (to reduce a certain kind of error to the optimal batch joint solution).

---

### Review · Reviewer_Njsb · 2023-08-17

**Summary Of Contributions:**

The paper studies the combination of functional regularization and weight regularization strategies in continual learning, under the lens of K-priors introduced in (Khan and Swaroop, 2021). A K-prior is a combination of a weight and functional regulariser used to adapt one model from one scenario to another (eg, adding data, changing regularizer, etc.), and it is defined to faithfully approximate the gradient on the past scenario.

The authors extend (Khan and Swaroop, 2021) by focusing on neural networks and on CL, where the network is adapted repeatedly to multiple tasks. In this case, the K-prior has two sources of errors, the limited memory of the past and the NN error. They improve on this by considering an EWC-style term for the former (stemming from a 1st order approximation of the K-prior) and an experience replay term on a second memory for the latter. Their final algorithm (Eq. (17)) is a sum of EWC, experience replay, functional regularization, and L2 regularization on the weights.

They evaluate their algorithm on several task-incremental learning (TIL) datasets with a few ablations and comparisons, showing that the method performs well.

**Audience:**

Yes

**Broader Impact Concerns:**

NA.

**Claims And Evidence:**

Yes

**Requested Changes:**

1. Please clarify more carefully which derivations in the paper are novel and which are straightforward applications of the material from (Khan and Swaroop, 2021).
2. Ablate the use of separate memories in the algorithm (comparing to a single memory with double the size used in all terms).
3. Include benchmarks on CIL and extend the comparisons to state-of-the-art models.
4. Focus more the related works section.

**Strengths And Weaknesses:**

The paper is well written, easy to read, and most ideas can be grasped quickly (I especially commend the authors on their use of very precise notations throughout).

On the negative side, The authors claim to study how to "optimally combine [CL] approaches". However, the algorithm they propose (Eq. (17)) is very simple, as a sum of all the possible regularization terms. The only novelty is how the memory is structured (two sets of past data, with the EWC matrix computed on the remaining subset of data). Unless I am mistaken, this is not benchmarked in the experiments, eg, what happens if I ignore this and use double the memory for both terms?

Also note that the major difficulty in combining regularization terms is weighting them appropriately. Here the authors are simply assuming a weight of 1 for all of them, which turns out to be suboptimal in some scenarios (as they are forced to remove certain terms in some experiments). I think that saying they "solved" the issue is a bit of an overstatement.

The novelty compared to (Khan and Swaroop, 2021) is also a bit overstated. K-priors with limited memory are explored in Section 3.2  of (Khan and Swaroop, 2021) using precisely a first-order approximation (which is the same that is done in this paper), while the connection to experience replay and NNs is explored in Section 4.2 of (Khan and Swaroop, 2021).

Experiments are only done on TIL scenarios, which is a bit limiting nowadays. I would have expected at least 1 comparison in a CIL scenario. In addition, I am unsure why baselines (and not ablations) are only considered in a small subset of experiments (Fig. 3, right). Why not comparing on all datasets?

The paper should also discuss a bit more in-depth related works, such as GEM, functional regularization on the embeddings, FROMP, etc.

---

> ### Author Response · Authors · 2023-08-30
> **Response to Reviewer Njsb (Part 1/2)**
>
> > The authors claim to study how to "optimally combine [CL] approaches". However, the algorithm they propose (Eq. (17)) is very simple, as a sum of all the possible regularization terms. The only novelty is how the memory is structured (two sets of past data, with the EWC matrix computed on the remaining subset of data). Unless I am mistaken, this is not benchmarked in the experiments, eg, what happens if I ignore this and use double the memory for both terms? Requested change: Ablate the use of separate memories in the algorithm (comparing to a single memory with double the size used in all terms).
>
> We believe simplicity is a strength of our approach. We disagree that memory structure is the "only" novelty. Previous efforts to combine different CL methods are mostly based on ad-hoc approaches. Our paper fixes this by providing a theoretical basis and the memory structure naturally follows from it (and is not the main novelty). While the experiment you suggest might be interesting, it would not help validate our theory or offer useful insights about it, which is the focus of our paper.
>
> > Also note that the major difficulty in combining regularization terms is weighting them appropriately. Here the authors are simply assuming a weight of 1 for all of them, which turns out to be suboptimal in some scenarios (as they are forced to remove certain terms in some experiments). I think that saying they "solved" the issue is a bit of an overstatement.
>
> This is a misunderstanding: we do not simply assume a weight of 1. In fact, this follows from the principle of gradient reconstruction. One of our main contributions is to show this. We are not forced to remove the term, rather we find that the term did not improve performance for the Split-TinyImageNet experiment. As pointed out in footnote 4 in Section 6.3, we hypothesize that this is because the model achieves almost perfect training accuracy on all tasks, such that the neural network error term is close to zero and does not need to be corrected for.
>
> We agree that we did not “solve” the issue, but we also do not claim so in the paper (i.e., we do not use the word “solve” or a synonym thereof in this context – if you disagree, please let us know which claim exactly you are referring to, so that we can soften it). The closest we could find to such a claim is us saying that we “fix” several issues with K-priors in Section 4, which we have now softened; in particular, we made the following changes:
> - Section 4, second-to-last paragraph: "which we fix in this paper" –> "which we _address_ in this paper"
> - Section 4, last paragraph:  "In this paper, we will fix these issues" --> "In this paper, we will _address_ these issues"
>
>
> > The novelty compared to (Khan and Swaroop, 2021) is also a bit overstated. K-priors with limited memory are explored in Section 3.2 of (Khan and Swaroop, 2021) using precisely a first-order approximation (which is the same that is done in this paper), while the connection to experience replay and NNs is explored in Section 4.2 of (Khan and Swaroop, 2021). Requested change: Please clarify more carefully which derivations in the paper are novel and which are straightforward applications of the material from (Khan and Swaroop, 2021).
>
> You are right that we follow Khan and Swaroop (2021) for the equations/derivations you mentioned, which we have now clarified in the paper; in particular, we added the following statements:
> - Section 5.1, footnote 1: “Note that the combined error term in (9) unifies the limited memory error term and the neural network error term stated in, respectively, Eq. (11) and Eq. (16) in Khan & Swaroop (2021).”
> - Section 5.2, footnote 2: “Note that the derivation of (11) follows the derivation of Eq. (12) in Khan & Swaroop (2021).”
> - Appendix A, footnote 7: “Note that this derivation follows the derivation in Appendix C in Khan & Swaroop (2021).”
>
> In summary, our novelty over Khan and Swaroop (2021) is as follows (both in terms of methodology and experiments):
> 1. We use Khan and Swaroop's results to actually reduce the error terms (they did not do this, and we do not think that this is a straightforward application).
> 2. We consider the continual learning setting (they considered only 1-step adaptation settings)
> 3. We did large-scale experiments (they only did small-scale ones).
>
> For more details on points 2. and 3. above, see the last paragraph of our related work section (Section 2).

---

> > ### Author Response · Authors · 2023-08-30
> > **Response to Reviewer Njsb (Part 2/2)**
> >
> > > Experiments are only done on TIL scenarios, which is a bit limiting nowadays. I would have expected at least 1 comparison in a CIL scenario. Requested change: Include benchmarks on CIL.
> >
> > While we agree that adding an experiment in a CIL scenario would be interesting, we focused on TIL scenarios in this paper for simplicity, and plan to extend our method to other CL scenarios in future work. We clarified throughout the paper that we focus on TIL benchmarks exclusively, to avoid any (implicit) claims that our method also performs well in other CIL scenarios; in particular, we made the following changes:
> > - Abstract: “improves performance on standard benchmarks” → “improves performance on standard _task-incremental continual learning_ benchmarks”
> > - Section 2, third paragraph: we added the sentence “In this work, we exclusively focus on task-incremental learning.”
> > - Conclusion: “on the benchmarks considered.” → ”on the _task-incremental continual learning_ benchmarks considered.”
> >
> >
> > > I am unsure why baselines (and not ablations) are only considered in a small subset of experiments (Fig. 3, right). Why not compare all datasets? Requested change: Extend the comparisons to state-of-the-art models.
> >
> > The reason for not including those results beyond Split-TinyImageNet is that we did not implement or run the baselines in Figure 3 (right) ourselves, but instead took the numbers from De Lange et al. (2021). Note that we do consider the popular Online EWC baseline in all our experiments.
> >
> >
> > > The paper should also discuss a bit more in-depth related works, such as GEM, functional regularization on the embeddings, FROMP, etc. Requested change: Focus more on the related works section.
> >
> > Agreed, thanks for pointing this out. Following your suggestion, we added further discussion on the relevant works you mentioned; in particular, we added another paragraph to the related work section, see the second paragraph of Section 2.

---

> > > ### Comment · Reviewer_Njsb · 2023-08-31
> > >
> > > I thank the authors for their reply. I would like to address a few of their points below.
> > >
> > > > *Our paper fixes this by providing a theoretical basis and the memory structure naturally follows from it*
> > >
> > > First, I agree that it is nice to have a methodological basis for their algorithm, and I already stated in my review that it is a positive point of the paper. However, from the POV of a person using CL, the algorithm they obtain is relatively simple, and summing multiple terms in this fashion is something that is done routinely. Hence, if I **only** look at the practical side of the paper, I feel it is correct to say that the major novelty is how the memory ends up being structured. I see the theoretical insights and the practical insights to be separated, the paper excels in the former but ends up being low-key on the latter.
> > >
> > > > *While the experiment you suggest might be interesting, it would not help validate our theory or offer useful insights about it*
> > >
> > > I disagree, it would help to show that the method they propose is motivated also from an empirical POV. It is a standard ablation in this setting.
> > >
> > > > *We are not forced to remove the term*
> > >
> > > I apologize, I understood from reading that the term was removed.
> > >
> > > > *In summary, our novelty over Khan and Swaroop (2021) is [...]*
> > >
> > > I agree with this summary.
> > >
> > > > *While we agree that adding an experiment in a CIL scenario would be interesting*
> > >
> > > Papers entirely focused on TIL are rare nowadays, but I understand this point.

---

> > > > ### Author Response · Authors · 2023-08-31
> > > > **Response to Reviewer Njsb**
> > > >
> > > > Thank you very much for your quick response, and again for your thorough review, which helped us improve our paper!
> > > >
> > > > > I see the theoretical insights and the practical insights to be separated, the paper excels in the former but ends up being low-key on the latter.
> > > >
> > > > Indeed, the main focus of our paper is on the theoretical development of a new continual learning method, with experiments to corroborate our theory and empirically validate that our method works in certain settings (incl. on the large-scale  ImageNet-1000 benchmark, which isn’t often considered in theoretically-focused papers).
> > > > We agree with you that our empirical evaluation is far from exhaustive, and we plan to follow your suggestions for additional experiments in future work -- thank you again for sharing your ideas on this.
> > > >
> > > > Finally, note that our work is in stark contrast to most papers on continual learning, which are predominantly empirically-driven, often with zero or only very little theoretical insights on why the proposed method should work.
> > > > We thus believe that our paper would be a valuable addition to the continual learning research landscape.

---

### Comment · Action_Editors · 2023-08-17
**Discussion phase**

Dear Authors and dear Reviewers,

We have entered the discussion phase. Please get familiar with the reviews and do not hesitate to respond to them.
Let's have a fruitful discussion!

Best regards,
Your AC

---

### Author Response · Authors · 2023-08-30
**General response**

We thank Reviewers Njsb, ioCd, GymF for their detailed and insightful comments. We are pleased that the reviewers unanimously highlighted that our paper is clearly written and the exposition of our method is easy to understand (Reviewers Njsb, ioCd, GymF). We are also glad about the appreciation of our use of theoretically grounded arguments in combining different continual learning techniques (Reviewer GymF).

Reviewer GymF was mainly concerned about some of our claims regarding our method’s design not being sufficiently supported by evidence. We agree with some of this, and softened those claims accordingly to fulfill the acceptance criteria. We rebut the other points in responses to each reviewer, and uploaded a revision of our paper that incorporates the reviewer feedback.

---

### Decision · Action_Editors · 2023-10-05

**Recommendation:** Accept as is

**Comment:**

Overall, the paper is well-written and provides a theoretical foundation for methods in continual learning. There is some doubt about the novelty of the paper, however, it provides enough theoretical and empirical evidence for its claims.
The authors did a good job in the rebuttal, answering the concerns raised by the reviewers in a satisfactory manner.

**Audience:**

It is a relevant position for the audience of TMLR, especially researchers interested in continual learning.

**Claims And Evidence:**

The paper makes the following claims:
1) An introduction of a theoretically-motivated approach to combine and improve regularization and experience replay.
2) An extension of the K-prior to multiple tasks is proposed and it is used to combine different regularization and replay methods.
3) The proposed approach leads to consistent improvements on standard benchmarks for multi-task image classification in task-incremental continual learning, such as Split-CIFAR, Split-TinyImageNet, and ImageNet-1000, across various memory budgets from small to large sizes.

The reviewers agree that the paper provides evidence for the claims, however, there is some doubt about the novelty of the paper. Nevertheless, the paper provides what it promises.